# Anticoagulation strategies in critical care for the treatment of atrial fibrillation: a protocol for a systematic review and meta-analysis

Brian Johnston [1], Alexandra Nelson,[2] Alicia C Waite,[1] Gedeon Lemma,[2] Ingeborg Welters [1]

¹Institute of Ageing and Chronic Disease, University of Liverpool, Liverpool, UK
²Faculty of Health and Life Sciences, University of Liverpool, Liverpool, UK

**Correspondence to**
Dr Brian Johnston;
brian.johnston@liverpool.ac.uk

## ABSTRACT

**Introduction** Atrial fibrillation (AF) is the most common cardiac arrhythmia in critically ill patients and is associated with an increased risk of thromboembolic events and mortality. Oral anticoagulation for thromboembolism prophylaxis is a key component of managing AF in the general population and is recommended by National Institute for Health and Care Excellence guidelines. However, assessment tools used to aid decision making about anticoagulation have not yet been validated in the critical care setting. There is a paucity of data assessing the impact of anticoagulation strategies on clinical outcomes in critically ill patients with AF. We present a protocol for a systematic review and meta-analysis to evaluate the effectiveness of anticoagulation strategies for AF used specifically in critical care.

**Methods and analysis** We will conduct a systematic review of the literature by searching MEDLINE, EMBASE, CENTRAL and PubMed databases for articles published from January 1990 to October 2019. Studies reporting anticoagulation strategies for AF in adults (>18 years) admitted to a general critical care setting will be assessed for inclusion. Outcomes of interest will include (1) percentage of patients started on anticoagulation in critical care for AF, (2) incidence of thromboembolism, (3) incidence of bleeding events, (4) intensive care unit (ICU) mortality, (5) hospital mortality, (6) ICU length of stay and (7) hospital length of stay. We will conduct a meta-analysis of trials. Risk of bias will be assessed using the Cochrane Risk of Bias tool for randomised trials or the Newcastle-Ottawa Risk of Bias assessment tool for non-randomised studies. This protocol and subsequent systematic review will be reported following the Preferred Reporting Items for Systematic Reviews and Meta-Analyses checklist.

**Ethics and dissemination** This proposed systematic review will include data extracted from published studies; therefore, ethical approval is not required. The results of this review will be published in clinical specialty journals and presented at international meetings and conferences.

**Trial registration number** CRD42020158237.

### Strengths and limitations of this study

► This systematic review follows the Preferred Reporting Items for Systematic Review and Meta-Analysis Protocols guidelines.
► Study selection, data extraction and risk of bias assessment will be performed independently by two researchers, ensuring that all relevant studies are included without personal biases.
► There are no language publication restrictions to this systematic review eliminating language bias.
► Non-randomised studies will be included in this review potentiating a high risk of bias in included studies.
► Given the particular risk for thromboembolic events in critically ill patients, we will only include studies on patients receiving anticoagulation during their critical care admission.

population. AF is characterised by erratic conduction of electrical impulses and unco-ordinated contraction of the atria which ultimately increases the risk of heart failure, stroke and death.[1 2] The reported incidence of new-onset AF (NOAF) in critically unwell patients admitted to intensive care unit (ICU) has been estimated to be between 1.8% and 10%.[1] However, a more recent study reported NOAF in 418 of 1782 (23%) of patients admitted to ICU and was associated with increased hospital mortality.[2 3] AF can trigger rapid ventricular rates resulting in haemodynamic compromise and a loss of cardiac output. This may contribute to acute decompensation of already unstable critically ill patients and eventually lead to the increased morbidity and mortality associated with AF in critical care.[2 4]

In addition to the traditional risk factors for the development of AF, including advanced age, hypertension, ischaemic heart disease, heart failure and valvular disease that

## INTRODUCTION
### Background

Atrial fibrillation (AF) is the most common cardiac arrhythmia in the critical care

predispose to pre-existing AF (PEAF), there are factors related to being critically ill that predispose patients to the development of NOAF.[1 2] These factors include electrolyte abnormalities, hypoxaemia, adrenergic overstimulation, progressive autonomic dysfunction, acute systemic inflammation, sepsis and shock.[1 2] Changes in autonomic activity resulting from vasopressor administration as well as electrolyte disturbances are frequent among critically ill patients and can lead to increased atrial ectopic impulses and subsequent NOAF.[4] There is also evidence that central venous catheters induce mechanical irritation of the atria and may be a contributing factor in the development of NOAF.[1] The high incidence of NOAF in critical care is likely due to a combination of these traditional and critical illness specific related risk factors.[4]

Both PEAF and NOAF are associated with a myriad of complications in critical care. As a result of inefficient atrial systole, the reduced blood flow velocity in the left atria predisposes patients with AF to cardiac and systemic emboli.[5] Embolic events, such as ischaemic stroke, are common and disabling complications of AF and cause a significant disease burden.[6] AF in the critical care setting is associated with a twofold increased risk of stroke and a twofold to fivefold increased risk of mortality.[2 7]

Oral anticoagulation for thromboembolism prophylaxis is a key component of managing AF in the general population.[8] In the UK, clinicians are guided in the management of NOAF by the recommendations from the National Institute for Health and Care Excellence (NICE). With regard to anticoagulation for NOAF, NICE recommends the use of validated tools assessing thromboembolic risk (eg, $CHA_2DS_2$-VASc) and bleeding risk (eg, HASBLED) to stratify patients that may benefit from systemic anticoagulation through prevention of thromboembolic events such as stroke.[8] However, the risk-benefit tools used to aid decision making about anticoagulation, such as the $CHA_2DS_2$-VASc and HASBLED scores, have not been validated in critical care populations.[9 10] Decisions around anticoagulation strategies in critical care are complex and challenging. Critically unwell patients are at a significantly increased risk of bleeding, but may also of hypercoagulable, due to the abnormal haemostasis that is associated with critical illness.[10] Factors contributing to the acquired coagulopathy in critical care include severe sepsis, thrombocytopenia, haemodilution of clotting factors following blood transfusion, disseminated intravascular coagulation, acute kidney injury and liver failure.[11] The potential need for urgent procedures or insertion of invasive devices, such as arterial lines and central venous catheters, poses an additional challenge in effectively anticoagulating these patients.[12] Combined with the added effect of anticoagulation itself, the high risk of bleeding may preclude safe therapeutic anticoagulation for AF in critical care patients. Guidelines from NICE recommend anticoagulation with Heparin in patients presenting with NOAF that are receiving no or subtherapeutic anticoagulation. Furthermore, they recommend that patients who have failed to achieve stable sinus rhythm within 48 hours or have risk factors for the recurrence of NOAF should be offered oral anticoagulation

long term. Despite these recommendations, a nationwide survey of intensive care clinicians revealed that 63.8% of clinicians would not routinely anticoagulate critically ill patients with NOAF while 30.8% would consider anticoagulation if NOAF persisted beyond 72 hours rather than the recommended 48 hours by NICE.[13] Furthermore, 98% of critical care clinicians revealed that they would be happy to administer Heparin as anticoagulation but not an oral anticoagulation.[13] The variation in practice likely represents the unique challenges of managing NOAF in critically unwell patients, in which administration of oral anticoagulants may not possible in sedated patients or via nasogastric tube and highlights the need for guidelines specific to critically ill patients.

### Why is it important to undertake this review?

There is a paucity of data assessing the impact of anticoagulation strategies on clinical outcomes in critically unwell patients with AF.[14] A consensus on an effective anticoagulation strategy for thromboembolism prophylaxis has not yet been reached and current recommendations regarding anticoagulation therapy are largely based on observational studies and expert opinion.[4] We therefore have designed a protocol for a systematic review and meta-analysis of the literature around the anticoagulation for AF in the critically unwell and critical care setting. We will assess the existing literature to define anticoagulation strategies in critical care for both NOAF and PEAF. We anticipate that the results of this review will highlight areas where evidence is lacking, trigger further research and contribute to the development of new guidelines specific to the management of anticoagulation in patients with AF in the critical care setting.

### Objective

To conduct a systematic review and meta-analysis of the literature to determine the use and effectiveness of anticoagulation strategies for AF in critical care and identify the associated risks and benefits of therapeutic anticoagulation for AF.

### METHODS AND ANALYSIS

This systematic review will be conducted in accordance with The Cochrane Collaboration principles of Systematic Reviews and will be reported following the Preferred Reporting Items for Systematic Reviews and Meta-Analyses (PRISMA) guidelines.[15 16]

This protocol has been registered with the International Prospective Register of Systematic Reviews (PROSPERO) database. PROSPERO registration number: CRD42020158237

### Inclusion criteria
#### Type of studies

All quantitative studies that report anticoagulation strategies for AF, in an adult critical care setting, will be assessed for inclusion. Eligible studies will include randomised and non-randomised studies.

Eligible studies must include both a cohort of patients with AF who were anticoagulated and a cohort of patients with AF who were not anticoagulated. We will include studies in patients diagnosed with NOAF (including paroxysmal AF) or PEAF (including permanent AF) by rhythm classification by continuous ECG monitoring or 12 lead ECG.

We will include studies conducted in level 2 and level 3 critical care settings as defined by Marshall *et al*.[17] We will include studies enrolling patients from general medical, general surgical or mixed mixed/surgical patients.

### Phenomenon of interest

Studies must include patients who have been in AF for >48 hours, based on the NICE guidelines on initiating anticoagulant treatment[8] and may include both PEAF and NOAF. Studies must describe outcomes associated with the chosen anticoagulation strategy including the outcomes of interest described below.

### Population

Studies that include adult patients admitted to the ICU types specified above will be assessed for inclusion. For the purpose of this review, an adult is defined as ≥18 years.

### Type of intervention and comparator(s)
#### Interventions

Eligible studies will include any of the following treatments:
1. Warfarin.
2. Any other vitamin K antagonist.
3. Unfractionated heparin (UFH).
4. Treatment dose low molecular weight heparin (LMWH).
5. Factor Xa inhibitors (eg, Edoxaban, Apixaban, Rivaroxaban).
6. Direct thrombin (Factor IIa) inhibitors (Dabigatran, Inogatran, Melagatran, Argatroban).
7. Any combination of the above interventions.

#### Comparators

Comparators will include:
1. Any of the interventions above.
2. Placebo.
3. Standard care where it does not include anticoagulation.

### Exclusion criteria
#### Types of studies

Qualitative studies, case studies, editorials, letters, abstract only reports, reviews and commentaries that do not include original information will be excluded from this review.

Studies of cohorts that have undergone or plan to undergo cardiothoracic surgery, permanent pacemaker insertion or surgical ablation will be excluded. Studies based on service-specific ICUs, for example, cardiac, cardiothoracic surgical or neurosurgical units, will be excluded in addition to studies based on acute medical units or in emergency departments.

### Phenomenon of interest

Studies will be excluded if patients have been started on any anticoagulation therapy for a reason other than AF and cannot be disaggregated from the entire cohort. Studies that include patients with an inherited or pre-existing bleeding disorder or clotting disorder will also be excluded.

### Population

Studies of participants under 18 years, pregnant women and patients with clear contraindication to anticoagulation therapy, for example, intracranial haemorrhage, overt bleeding or allergy to anticoagulant medication will be excluded.

### Outcome measures
#### Primary outcome measure

1. Percentage of patients started on anticoagulation for AF (NOAF and/or PEAF) in a critical care setting including but not limited to warfarin, LMWH, UFH, Factor Xa inhibitors, Factor IIa inhibitors.

#### Secondary outcome measures

1. Incidence of thromboembolic events (defined as stroke, mesenteric ischaemia, acute limb ischaemia, pulmonary embolism) during critical care admission to identify the outcomes of anticoagulation therapy in the critical care setting.
2. Incidence of any in hospital thromboembolic events (defined as stroke, mesenteric ischaemia, acute limb ischaemia, pulmonary embolism).
3. Development of a major haemorrhage event (defined clinically as haemorrhage leading to death, signs of shock, requirement for blood transfusion or urgent endoscopic / surgical intervention).[18]
4. ICU length of stay (LOS).
5. ICU mortality.
6. 28-day mortality.
7. 90-day mortality.
8. 365-day mortality.
9. Use of risk stratification scores for anticoagulation decision making (such as $CHA_2DS_2$-VASc and HASBLED).
10. Severity scores of critical illness (eg, APACHE II and SOFA).

### Search strategy

We will engage the services of a health information specialist, and a comprehensive broad literature search will be conducted. Medical subject headings will be used to identify papers that matched the search index and relevant selected free text terms will also be used.

The search strategies identified will focus on the population (critically ill), the participants (patients with AF) and the intervention (any anticoagulation strategy mentioned previously). The search strategy will not be limited by outcomes studied in order to broaden the scope of eligible papers.

## Database searches

We will search the following databases for studies published between January 1990 and October 2019;

1. MEDLINE Ovid, Medline Ovid Epub Ahead of Print, and Medline Ovid In-Process.
2. EMBASE Ovid.
3. The Cochrane Central Register of Controlled Trials.
4. PUBMED.
5. ClinicalTrials.gov.

MEDLINE, EMBASE and PUBMED will be access via NICE Healthcare Database Advanced Search (HDAS) using OpenAthens. We will screen referenced papers following full text screening and any review articles identified during the screening process. We will screen for relevant conference proceedings and include any potentially relevant studies. A full description of the search strategy that will be used in HDAS (online supplemental file).

## Citation management and screening

Results from searches in all databases will be exported to Endnote X9 (Clarivate analytics) and duplicates will be removed. All citations will be imported into the Covidence systematic review platform (Veritas Health Innovation, Melbourne, Australia).[19] Titles and abstracts will be screened in duplicate by two independent reviewers for relevance to the review and eliminated as appropriate. Full text articles that are not excluded after title and abstract screening will be screened against the inclusion and exclusion criteria by two independent reviewers.

Any discrepancies or conflicts in the screening process will be resolved by discussion and subsequent input of a third reviewer. Primary reason for exclusion will be documented in Covidence and the screening process will be documented in a PRISMA flowchart (figure 1).

## Data extraction and management

Data will be extracted in duplicate by two independent reviewers using a standardised, prepiloted data extraction form.

Information, including the following characteristics, will be extracted from studies:

1. Study design and methodology, including title, authors, journal, publication date, study type, study period and number of participants.
2. All inclusion criteria previously mentioned.
3. Population characteristics, including age, sex, ICU admission diagnosis, location, co-morbidities, illness severity scores.
4. Recruitment procedures.
5. Interventions.
6. The primary and secondary outcome measures and reported findings of the study.

Any discrepancies within data extraction will be resolved through discussion and consultation with a third reviewer. We will contact authors if clarification regarding the data or methodology is required. If data cannot be obtained from the authors, the impact will be discussed as a limitation of the review.

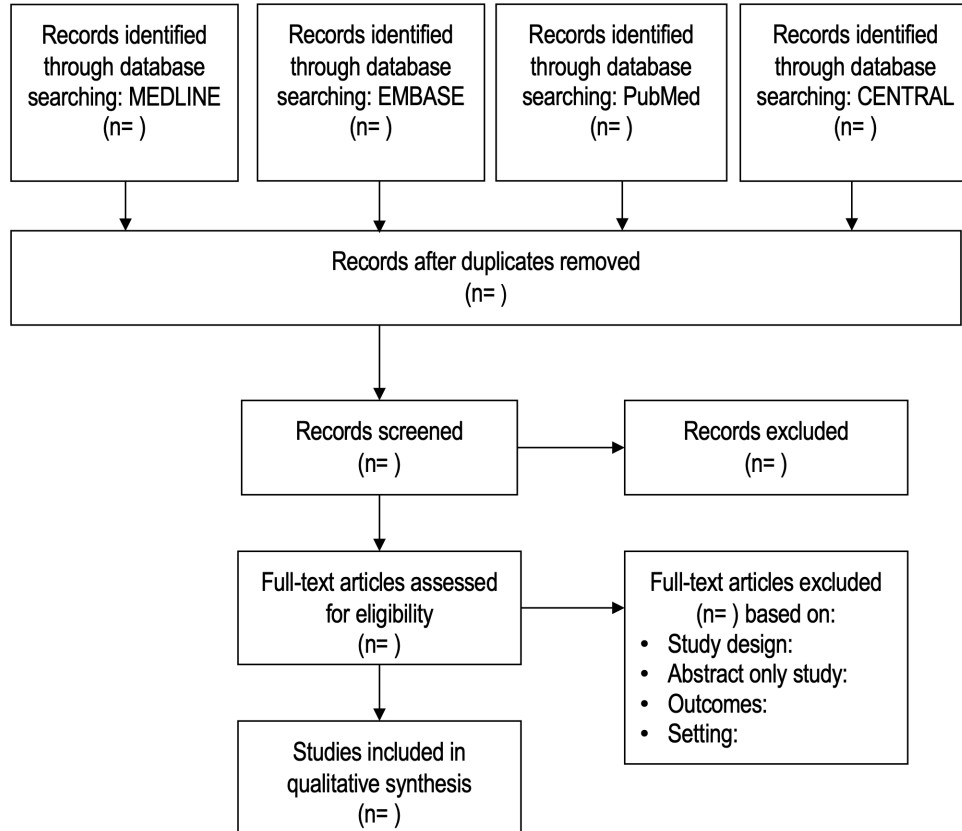

**Figure 1** PRISMA flowchart of studies selected in the systematic review.

## Risk of bias assessment

Risk of bias will be assessed using a modified Newcastle-Ottawa Scale (mNOS), a scoring system for non-randomised trials.[20] mNOS uses a system by which the study is judged on three broad perspectives: the selection of the study groups, the comparability and the ascertainment of outcome of interest.

Included randomised controlled trials will be assessed with the revised Cochrane Collaborations Risk of Bias (RoB 2) tool using the criteria outlined in the Cochrane Handbook of Systematic Reviews of Interventions.[21] The RoB 2 tool assesses the following domains: selection, comparability, outcome performance and will be presented in table format.

## Data synthesis and analysis

A description of included studies will be reported in evidence tables and discussed in the text. We will report participant characteristics, interventions, clinical outcomes and methodological quality.

Primary outcomes will be reported as the percentage of patients that received anticoagulation for any AF (NOAF and PEAF) in critical care. If sufficient data allow, we will report the percentage of patients started on anticoagulation according to type of AF (NOAF and/or PEAF).

We anticipate that the majority of studies included in our systematic review and meta-analysis will be case control studies. Therefore, we will present dichotomous data as number of participants experiencing the outcome with OR and 95% CIs.[22] For continuous outcome data, we will extract arithmetic means and SD with 95% CIs for each outcome, together with the numbers analysed in each group. We will also extract medians and ranges where provided.

Data from individual studies will be pooled and presented visually by Forest plot. We anticipate studies to vary significantly; therefore, we will undertake a meta-analysis of included studies by using a random effects model with 95% CI to estimate pooled estimates of effect. We will assess for heterogeneity between studies using the Cochran's Q test and report the degree of heterogeneity using the $I^2$ statistic in RevMan (Review Manager, 2014). Heterogeneity will be considered low if values are below 25%, moderated if values are between 25% and 75% and high if values are above 75%. Suitability for meta-analysis will be determined by the degree of clinical, statistical and methodological heterogeneity observed between studies. We will explore heterogeneity between studies by subgroup analysis assessing a number of factors such as study design, intervention and participants. Where there is a high degree of heterogeneity (>75%) that cannot be explained by clinical or methodological variation, we will not undertake a meta-analysis of results. If we are unable to pool studies due to heterogeneity, we will describe the findings via a narrative synthesis. Meta-analysis will be conducted using the Cochrane Collaborations RevMan (Review Manager, 2014) software.

We will assess the certainty of our evidence for each outcome according to the Grading of Recommendations Assessment, Development and Evaluation (GRADE) approach.[23] GRADE methodology assesses the certainty in evidence across the domains of risk of bias, consistency, directness, precision and publication bias. The certainty of evidence will be reported as high, moderate, low or very low.

## Subgroup analysis

If we extract sufficient data, we will undertake subgroup analysis, assessing the following:
1. Type of anticoagulant therapy.
2. NOAF versus PEAF.
3. Sepsis versus non-sepsis.
4. Illness severity scores (APACHE II score, SOFA score).
5. Risk scores for bleeding and thromboembolic events (HASBLED, $CHA_2DS_2$-VASc, respectively).
6. Medical versus surgical ICU admission diagnosis.

## Patient and public involvement

This protocol will use previously published data. As such, there will be no patient and public involvement in the design of the study, interpretation of the results. We will disseminate the results of this review to any interested patients and it will be freely accessible in open access journals.

## ETHICS AND DISSEMINATION

This proposed systematic review will not include primary data and will extract data from published studies; therefore, ethical approval is not required. The results of this review will be disseminated through publication in clinical specialty journals, including open access journals, and various medias including international meetings, conferences, congresses and symposiums.

## DISCUSSION

This systematic review will assess the evidence available for anticoagulation strategies used in the management of AF in critically ill patients. AF affects up to 25% of critical care admissions, hence the need for systematic evaluation of anticoagulation strategies used in critical care and their impact on patient outcomes.[24] AF is associated with an increased risk of thromboembolic complications, but scoring systems used to aid decision making on whether and how-to anticoagulate patients with AF have not been validated for use in the critical care setting. This review will enable a comparison of the anticoagulation strategies used for both PEAF and NOAF in critical care and aims to assess outcomes including thromboembolic events, haemorrhagic events, LOS in critical care and mortality on ICU. Critically ill patients have a particularly high risk of thromboembolic events, which could theoretically be ameliorated by administering anticoagulant therapy during critical illness. A limitation of our protocol is that

results will not be generalisable to patients who are clinically stable after having been critically ill.

Based on a preliminary search of the literature, we believe that this is the first systematic review to consider anticoagulation strategies used in the treatment of AF in the critically ill patient cohort. We aim for this review to provide an evidence base on which recommendations can be made, to ultimately reduce the impact of AF on patient outcomes during and after critical care admission. We will publish the results of this review in clinical specialty journals and we present it at international meetings and conferences. We will also make our report available via our web- based repository and update our PROSPERO entry with the web address details.

**Contributors** The protocol was conceived and designed by IW, BJ and ACW. AN will conduct primary screening and data collection, reviewed by BJ and GL. Data extraction, analysis and preparation of the manuscript will be conducted by AN and BJ. BJ and AN drafted the current protocol. IW, BJ, ACW, GL and AN will read and approve the final manuscript. All authors have read and approve the final manuscript.

**Funding** The authors have not declared a specific grant for this research from any funding agency in the public, commercial or not-for-profit sectors.

**Competing interests** None declared.

**Patient consent for publication** Not required.

**Provenance and peer review** Not commissioned; externally peer reviewed.

**ORCID iDs**
Brian Johnston http://orcid.org/0000-0003-1634-3297
Ingeborg Welters http://orcid.org/0000-0002-3408-8798

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
