## [Reviewer comments · BMJ Open]

ARTICLE DETAILS

TITLE (PROVISIONAL)	Anticoagulation strategies in critical care for the treatment of atrial fibrillation: a protocol for a systematic review and meta-analysis
AUTHORS	Johnston, Brian; Nelson, Alexandra; Waite, Alicia; Lemma, Gedeon; Welters, Ingeborg

VERSION 1 – REVIEW

REVIEWER	Ahmed AITurki McGill University
REVIEW RETURNED	11-Apr-2020

GENERAL COMMENTS	Very well written protocol for a systematic review. No issues identified.
---

REVIEWER	Mauricio Pimentel Hospital de Clínicas de Porto Alegre Brazil
REVIEW RETURNED	28-Apr-2020

GENERAL COMMENTS	The study protocol addresses the very important clinical issue of atrial fibrillation in critically ill patients. The protocol is well designed to answer its objectives. The authors could clarify why they planned to include patients who have been in AF > 48h and not > 24h.
---

REVIEWER	Heather Smith University of Ottawa, Canada
REVIEW RETURNED	19-May-2020

GENERAL COMMENTS	This protocol outlines the plans for a systematic review and meta-analysis to evaluate the effectiveness of anticoagulation strategies for atrial fibrillation in critical care. Overall it is well written and comprehensive. As described by the authors, there is a paucity of literature to guide anticoagulation therapy for AF in critical care, therefore this review could provide important clinical and academic contributions. My main criticism with this systematic review is that the outlined study does not compare the “effectiveness” of strategies for anticoagulation. Rather, it compares the rate of anticoagulation for different anticoagulation agents employed for AF among critically ill patients. If the role of oral anticoagulation is thromboembolism prophylaxis, and authors aim to compare the effectiveness of anticoagulation strategies then I would argue that the outcomes and comparators need to be revised to reflect this. Please see below additional minor revision recommendations: INTRODUCTION
---

	1. If there are well-defined strategies to guide anticoagulation therapy for AF in other patient cohorts for instance, cardiac surgery, I would add this to introduction. ABSTRACT 1. Line 41: recommend to change “inception” to a specific start date to provide a clear time frame for the planned review. METHODS 1. Line 143: recommend to clarify definition of included type of ICU. For instance, it is not clear what level of care is provided in a high dependency unit. Adopting a standardized definition such as the one published by Marshall et al, 2016, J Critical Care (doi: 10.1016/j.jcrc.2016.07.01). 2. Line 145: clarify definition of AF and rationale for 48hour cut-off. 3. Line 151: The sole exclusion criteria of age <18 and pregnancy could lead to important confounders. For instance, I would recommend excluding individuals with an allergy to the anticoagulation therapy of choice, or clear contraindication to anticoagulation (ie overt bleeding). 4. Line 192: Please provide justification for limiting the incidence of thromboembolic events only the patient’s admission to critical care. 5. Line 194: Provide definition for “major haemorrhage”. 6. Line 226: Clarify the years for study inclusion. For instance, “same search strategy to identify any studies that are published between commencing” suggests that studies published in 2020 will be included. 7. Add reference for Covidence. DISCUSSION 1. Line 308: if available, please provide reference for initial scoping review.
--	---

REVIEWER	Conrad Kabali Division of Epidemiology, Dalla Lana School of Puyblic Health, University of Toronto
REVIEW RETURNED	13-Jun-2020

GENERAL COMMENTS	The authors wrote a protocol for a systematic review and meta-analysis on anticoagulation strategies in critical care for the treatment of atrial fibrillation. Overall, the paper is well written and addresses an important research question. I have minor comments. Page 2, Line 45: The sentence “if sufficient data allows, we will conduct a meta-analysis” needs re-phrasing for clarity. Sufficient data could mean imprecision, something that a meta-analysis is trying to solve, by pooling data across studies. Did you want to say that a meta-analysis will be done if the assumption required to conduct it are met? Page 3, Line 58: The bullets are not talking about Strengths and Limitations. Please revise for consistency with the heading. Page5, Line 137: It is sufficient to say that eligible studies will include randomized and non-randomized studies. Page 10, Line 261: Cochrane has revised the RoB tool for randomized studies. Please consider using the most recent version of this tool.
---

	Page 11, Line 272: Why pick OR over RR, RD, or HR? This needs clarification. Page 11, Line 277: To be more specific please replace “chi-square test” with “Cochran’s Q test”. Page 11, Line 277: I2 is not an absolute measure of heterogeneity. You might want to read Borenstein et al. “Basics of meta-analysis: I2 is not an absolute measure of heterogeneity” to understand the reason, and revise how you will want to accurately evaluate heterogeneity. Page 11, Lines 278-281: Difference in effect sizes is a form of heterogeneity. When using a random effects meta analysis you are implicitly assuming that heterogeneity is present. If homogeneity assumptions are not violated what you need is the fixed effects approach.
--	---

VERSION 1 – AUTHOR RESPONSE

Reviewer comments

The authors could clarify why they planned to include patients who have been in AF > 48h and not > 24hr

The aim of our study is to assess the use and effectiveness of anticoagulation following the development of acute atrial fibrillation in critically unwell patients. Acute onset atrial fibrillation has been shown to most often develop in critically unwell patient within 72 hours of admission (Wetterslev, 2019). Current guidance on whether to commence anticoagulation in patients with atrial fibrillation depends upon risk assessment using validated tools such as CHADsVASc.(NICE 2014) However, these tools have not been validated for use in critically unwell patients. The authors have previously investigated intensive care clinician’s management of AF and found that 68% would not routinely anticoagulate patients that develop atrial fibrillation, whilst 30% would anticoagulate those that remained in atrial fibrillation after 72 hours(Chean, 2017). NICE guidance recommends that patients who develop acute onset atrial fibrillation should be offered anticoagulation if a stable sinus rhythm has not be establishing within the 48 hour period following onset (NICE, 2014). We therefore chose 48 hours based on the recommendations as per NICE guidelines and our experience of clinician practice within the United Kingdom.

- 1) Wetterslev et al. 2019. New onset atrial fibrillation in adult critically ill patients: a scoping review. *Intensive Care Med*;45:928-938
- 2) National Institute for Health and Care Excellence. Clinical Guideline CG180. Atrial Fibrillation Management. Online. Last accessed 09/07/2020: <https://www.nice.org.uk/guidance/cg180/resources/atrial-fibrillation-management-pdf-35109805981381>
- 3) Chean et al. 2017. Current practice in the management of new onset atrial fibrillation in critically ill patients: a UK-wide survey. *PeerJ*:5e3716; DOI 10.7717/peerj.3716

My main criticism with this systematic review is that the outlined study does not compare the “effectiveness” of strategies for anticoagulation. Rather, it compares the rate of anticoagulation for different anticoagulation agents employed for AF among critically ill patients. If the role of oral anticoagulation is thromboembolism prophylaxis, and authors aim to compare the effectiveness of anticoagulation strategies then I would argue that the outcomes and comparators need to be revised to reflect this.

We thank the reviewer for the comments but draw attention to the secondary objectives. We have previously conducted a nationwide survey of intensive care clinicians (Chean 2017). This survey revealed that only 30% intensive care clinicians would routinely anticoagulate patients who develop atrial fibrillation in critical care. This is despite NICE guidance suggesting that patients who develop new onset AF are anticoagulated if sinus rhythm has not been restored within 48 hours. Due to this considerable variation in clinical practice we feel that assessing the use/incidence of patients undergoing anticoagulation for new onset atrial fibrillation is a valuable outcome measure. In the secondary objectives we aim to compare the effectiveness of anticoagulation on prevention of thromboembolism and side effects such as major bleeding events. We have updated the primary outcome of our study to make this clear that we are assessing both the use of anticoagulation, its effectiveness and complications associated with anticoagulation use. Line 136 Page 5

- 1) Chean et al. 2017. Current practice in the management of new onset atrial fibrillation in critically ill patients: a UK-wide survey. PeerJ:5e3716; DOI 10.7717/peerj.3716

INTRODUCTION

If there are well-defined strategies to guide anticoagulation therapy for AF in other patient cohorts for instance, cardiac surgery, I would add this to introduction.

While the authors are aware of well-defined strategies for anticoagulation in other cohorts such as cardiac surgery, we feel that these patient cohorts are distinct both in risk factors and aetiology of the development of AF. (Bedford, 2019) The European Society of Cardiology guidelines differentiate AF that develops post-operatively and following cardiac or valvular surgery from AF developing in acute unwell patients (Kirchhof 2016). Furthermore, the indications and contra-indications to anticoagulation in patients undergoing cardiac surgery or post-operatively are likely to be surgery and context specific. The current study is aimed at patients developing atrial fibrillation due to acute illness rather than cardiac surgery or post-operatively in whom the primary indication for anticoagulation will be prevention of thromboembolic events. We feel that including post-operative and cardiac surgery patients will be out with the scope of this review and introduce a cohort of patients, separate to those which we are interested in.

- 1) Bedford et al. 2019 Risk factors for new-onset atrial fibrillation on the general adult ICU: A systematic Review. Journal of Critical Care;53:169-175

- 2) Kirchhof et al. 2016. 2016 ESC Guidelines for the management of atrial fibrillation developed in collaboration with EACTS. *European Heart Journal*;37(38):2893-2962

ABSTRACT

Line 41: recommend to change “inception” to a specific start date to provide a clear time frame for the planned review.

These dates have now been included in the main manuscript and abstract. We no longer refer to inception. Line 37, Page 2

METHODS

Line 143: recommend to clarify definition of included type of ICU. For instance, it is not clear what level of care is provided in a high dependency unit. Adopting a standardized definition such as the one published by Marshall et al, 2016, J Critical Care (doi: 10.1016/j.jcrc.2016.07.01).

We have updated this section of the manuscript to better clarify the type of ICU that we are including in our systematic review. We have agreed to adopt the proposed classification provided by Marshall et al to aid clarity. In this systematic review we will include Level 2 and Level 3 areas of care. Line 157. Page 6

Line 145: clarify definition of AF and rationale for 48hour cut-off.

We have included a definition of AF as per NICE guidelines 2014. Line 162. Page 6

With regard to the 48 hour cut off please see comment in answer to question above:

The aim of our study is to assess the use and effectiveness of anticoagulation following the development of acute atrial fibrillation in critically unwell patients. Acute onset atrial fibrillation has been shown to most often develop in critically unwell patient within 72 hours of admission (Wetterslev, 2019). Current guidance on whether to commence anticoagulation in patients with atrial fibrillation depends upon risk assessment using validated tools such as CHADsVASc.(NICE 2014) However, these tools have not been validated for use in critically unwell patients. The authors have previously investigated intensive care clinician’s management of AF and found that 68% would not routinely anticoagulate patients that develop atrial fibrillation, whilst 30% would anticoagulate those that remained in atrial fibrillation after 72 hours (Chean, 2017). NICE guidance recommends that patients who develop acute onset atrial fibrillation should be offered anticoagulation if a stable sinus rhythm has not be establishing within the 48 hour period following onset (NICE, 2014). We therefore chose 48 hours based on the recommendations as per NICE guidelines and our experience of clinician practice within the United Kingdom.

- 1) Wettestev et al. 2019. New onset atrial fibrillation in adult critically ill patients: a scoping review. *Intensive Care Med*;45:928-938
- 2) National Institute for Health and Care Excellence. Clinical Guideline CG180. Atrial Fibrillation Management. Online. Last accessed 09/07/2020: <https://www.nice.org.uk/guidance/cg180/resources/atrial-fibrillation-management-pdf-35109805981381>
- 3) Chean et al. 2017. Current practice in the management of new onset atrial fibrillation in critically ill patients: a UK-wide survey. *PeerJ*:5e3716; DOI 10.7717/peerj.3716

Line 151: The sole exclusion criteria of age <18 and pregnancy could lead to important confounders. For instance, I would recommend excluding individuals with an allergy to the anticoagulation therapy of choice, or clear contraindication to anticoagulation (ie overt bleeding).

We have updated our manuscript to include these as exclusion criteria. Line 199. Page 8.

Line 192: Please provide justification for limiting the incidence of thromboembolic events only the patient's admission to critical care.

We have limited the incidence of thromboembolic events to admission to critical care as this is our primary area of interest. Patients admitted to intensive care have a high incidence of thromboembolic events over and above ward level patients. It is estimated that thromboembolic events occur in between 5%-37% of admissions (Zhang et al. 2019). However critically unwell patients are also at increased risk of bleeding complications related to factors specific to critical illness such as coagulopathy due to systemic inflammation, shock, blood products, and organ failure. When deciding on anticoagulation strategies clinicians need to make a risk benefit judgement balancing the risk of thromboembolic events against that of bleeding events. This is a dynamic process during critical illness but it is likely that relative benefit of anticoagulation increases and the risk of bleeding decreases as patients recover and they are discharged from intensive care. Our population of interest are critically unwell patients therefore we have limited the incidence of thromboembolic events to those occurring during critically care admission as patients discharged from critically have a significantly different risk profile.

We accept that this may be a limitation as we will only include VTE in critical care. However, as stated these patients are well known to have a particularly high risk over and above ward level patients. To address this concern we have included, 'any in hospital thromboembolic event' if this data is available in the study. Line 214. Page 8.

Line 194: Provide definition for "major haemorrhage".

We have now included the definition for major haemorrhage. We have chosen a pragmatic clinical definition similar to that used in previous clinical research trials such as the HALT-IT trial.

- 1) HALT-IT Trial Coordinators. 2020 Effects of a high-dose 24-h infusion of tranexamic acid on death and thromboembolic events in patients with acute gastrointestinal bleeding (HALT-IT): an international randomised, double-blind, placebo-controlled trial. *Lancet*; 395; 1927-36

Line 226: Clarify the years for study inclusion. For instance, “same search strategy to identify any studies that are published between commencing” suggests that studies published in 2020 will be included.

This sentence has been removed and we have clarified the search strategy dates. Between January 1990 and October 2019.

Add reference for Covidence.

This has been included in the main publication using the recommended citation for Covidence. However, we have included the reference in the main references with website included.

DISCUSSION

Line 308: if available, please provide reference for initial scoping review.

We have changed the wording in the main manuscript to reflect that this was not a scoping review but was preliminary searches prior to this protocol. Line 344. Page 13

Page 2, Line 45: The sentence “if sufficient data allows, we will conduct a meta-analysis” needs re-phrasing for clarity. Sufficient data could mean imprecision, something that a meta-analysis is trying to solve, by pooling data across studies. Did you want to say that a meta-analysis will be done if the assumption required to conduct it are met?

We have clarified this in our manuscript and removed the sentence, ‘if sufficient data allows, we will conduct a meta-analysis.’ Line 42, Page 2.

Page 3, Line 58: The bullets are not talking about Strengths and Limitations. Please revise for consistency with the heading.

We have revised this in the main manuscript. Line 55. Page 3

Page5, Line 137: It is sufficient to say that eligible studies will include randomized and non-randomized studies.

We have altered this in the main manuscript under ‘*Types of studies*’ section. Line 152. Page 6.

Page 10, Line 261: Cochrane has revised the RoB tool for randomized studies. Please consider using the most recent version of this tool.

We have updated this area of the manuscript and intend to use the revised RoB tool for randomised studies. Line 278. Page 11.

Page 11, Line 272: Why pick OR over RR, RD, or HR? This needs clarification.

We have conducted a preliminary search of the available literature we anticipate that the majority of studies eligible for inclusion in our systematic review and meta-analysis will be retrospective care control studies. With this in mind we intend to present pooled estimates of effect using OR (Higgins et al 2019). We have included this in the main manuscript to clarify. Line 289. Page 11.

- 1) Higgins JPT, Li T, Deeks JJ. 2019 Chapter 6: Choosing effect measures and computing estimates of effect. In: Higgins JPT, Thomas J, Chandler J, Cumpston M, Li T, Page MJ, Welch VA. *Cochrane Handbook for Systematic Reviews of Interventions* version 6.0 (updated July 2019). Cochrane, 2019. Available from www.training.cochrane.org/handbook.

Page 11, Line 277: To be more specific please replace “chi-square test” with “Cochran’s Q test”.

We have clarified this in the main manuscript and replaced chi-squared with Cochran’s Q test. Line 297. Page 12.

Page 11, Line 277: I2 is not an absolute measure of heterogeneity. You might want to read Borenstein et al. “Basics of meta-analysis: I2 is not an absolute measure of heterogeneity” to understand the reason, and revise how you will want to accurately evaluate heterogeneity.

We have revised this area of our manuscript to make it clear how we intend to evaluate heterogeneity.

Page 11, Lines 278-281: Difference in effect sizes is a form of heterogeneity. When using a random effects meta analysis you are implicitly assuming that heterogeneity is present. If homogeneity assumptions are not violated what you need is the fixed effects approach.

We have revised this area of our manuscript. We anticipate that our studies will be subject to significant variation between studies. As such we will undertake a random effect model. We will assess for clinical, statistical and methodological heterogeneity. We will undertake subgroup analysis. Line 294 – 301. Page 12.

VERSION 2 – REVIEW

REVIEWER	Heather Smith University of Ottawa, Canada
REVIEW RETURNED	15-Jul-2020

GENERAL COMMENTS	The authors have provided significant revisions to the reviewed manuscript and addressed all of the reviewer comments. A few minor suggestions to the authors: Line 73-74: the authors state that 2/3 of patients develop NOAF. From the sentence structure it is not clear if this is referring to patients with PEAf or all ICU patients. Further, in the article referenced, <1/3 develop NOAF "In one study of 1,782 patients admitted to the ICU with sepsis, 418 (23%) developed new-onset AF; the new-onset AF was associated with increased hospital mortality after accounting for competing risks and multiple, time-varying confounding variables (subdistribution hazard ratio, 2.10 [95% CI, 1.61-2.73]).¹⁹" Please clarify and reference appropriately. Line 43: "Ottawa" This study is aimed at patients developing AF due to acute illness (not after surgery), however, it will include surgical patients. The authors have planned a subgroup analysis to compare surgical and medical critical care patients. I would recommend the authors, where possible, consider studies which exclude patients who are in the immediate post-operative period to avoid confounding incidence of post-operative atrial fibrillation. I will leave this to the discretion of the authors.
--

REVIEWER	Conrad Kabali Division of Epidemiology, Dalla Lana School of Public Health, University of Toronto
REVIEW RETURNED	15-Aug-2020

GENERAL COMMENTS	The authors have addressed all the comments raised. I have no further comments.
---

VERSION 2 – AUTHOR RESPONSE

Reviewer: 3
Reviewer Name
Heather Smith

Institution and Country
University of Ottawa, Canada

Please state any competing interests or state 'None declared':
None declared

Please leave your comments for the authors below

The authors have provided significant revisions to the reviewed manuscript and addressed all of the reviewer comments. A few minor suggestions to the authors:

Line 73-74: the authors state that 2/3 of patients develop NOAF. From the sentence structure it is not clear if this is referring to patients with PEAf or all ICU patients. Further, in the article referenced, <1/3 develop NOAF "In one study of 1,782 patients admitted to the ICU with sepsis, 418 (23%) developed new-onset AF; the new-onset AF was associated with increased hospital mortality after accounting for competing risks and multiple, time-varying cofounding variables (sub-distribution hazard ratio, 2.10 [95% CI, 1.61-2.73]).19 " Please clarify and reference appropriately.

We have rephrased this sentence and included an updated reference.

'The reported incidence of new-onset AF (NOAF) in critically unwell patients admitted to ICU has been estimated to be between 1.8% to 10% [1]. However, a more recent study reported NOAF in 418 of 1782 (23%) of patients admitted to ICU and was associated with increased hospital mortality [2][3]'

Line 43: "Ottawa"

This has been corrected. Thankyou.

This study is aimed at patients developing AF due to acute illness (not after surgery), however, it will include surgical patients. The authors have planned a subgroup analysis to compare surgical and medical critical care patients. I would recommend the authors, where possible, consider studies which exclude patients who are in the immediate post-operative period to avoid confounding incidence of post-operative atrial fibrillation. I will leave this to the discretion of the authors.

We thank-you for the comments. We will endeavour to ensure that any patients that we make it clear that any patients in the immediate post-operative period. The primary outcome of our study is whether or not patients with NOAF undergo anticoagulation. We anticipate that few patients will be anticoagulated in the immediate post-operative period. However, we will ensure that this is made clear in tables on the demographics of participants in studies in our write up.

Reviewer: 4

Reviewer Name

Conrad Kabali

Institution and Country

Division of Epidemiology, Dalla Lana School of Public Health, University of Toronto

Please state any competing interests or state 'None declared':

None declared

Please leave your comments for the authors below

The authors have addressed all the comments raised. I have no further comments.